# Reconstruction of the ancestral metazoan genome reveals an increase in genomic novelty

Jordi Paps[1,2] & Peter W.H. Holland [2]

Understanding the emergence of the Animal Kingdom is one of the major challenges of modern evolutionary biology. Many genomic changes took place along the evolutionary lineage that gave rise to the Metazoa. Recent research has revealed the role that co-option of old genes played during this transition, but the contribution of genomic novelty has not been fully assessed. Here, using extensive genome comparisons between metazoans and multiple outgroups, we infer the minimal protein-coding genome of the first animal, in addition to other eukaryotic ancestors, and estimate the proportion of novelties in these ancient genomes. Contrary to the prevailing view, this uncovers an unprecedented increase in the extent of genomic novelty during the origin of metazoans, and identifies 25 groups of metazoan-specific genes that are essential across the Animal Kingdom. We argue that internal genomic changes were as important as external factors in the emergence of animals.

---

[1] School of Biological Sciences, University of Essex, Colchester, Essex CO4 3SQ, UK. [2] Department of Zoology, University of Oxford, Oxford OX1 3PS, UK. Correspondence and requests for materials should be addressed to J.P. (email: jpapsm@essex.ac.uk)

**M**etazoa are the multicellular eukaryotic group with the largest number of described species, over 1.6 million[1]. They evolved within the eukaryotic supergroup Opisthokonta, most closely related to choanoflagellates, filastereans, and ichthyosporeans[2]. In addition to environmental and ecological triggers, biological functions encoded in the genome were crucial in this transition, including genes involved in differential gene regulation (e.g., several transcription factors, signalling pathways), cell adhesion (e.g., cadherins), cell type specification, cell cycle, and immunity[3]. Recent studies show that many genes typically associated with metazoan functions actually pre-date animals themselves[4], supporting functional co-option of 'unicellular genes' during the genesis of metazoans.

However, the role of genome novelty in animal origins has not been fully evaluated. We hypothesize that genomic novelty had a major impact in this transition, particularly involving biological functions which are hallmarks of animal multicellularity (gene regulation, signalling, cell adhesion, and cell cycle). Here we apply a comparative genomics approach using sophisticated methods, newly developed programs, and a comprehensive taxon sampling. The reconstruction of the ancestral genome of the last common ancestor of animals shows a set of biological functions similar to other eukaryote ancestors, while revealing an unexpected expansion of gene diversity. These analyses also highlight 25 groups of genes only found in animals that are highly retained in all their genomes, with essential functions linked to animal multicellularity.

## Results

**Inference of homology groups**. To perform a genome wide examination of novelty, we designed a bioinformatics pipeline to identify homologous groups of proteins within and between genomes, and determine their phylogenetic origin (Methods section and Supplementary Fig. 1). In contrast with other approaches (e.g., phylostratigraphy[5,6]), we use a large and representative taxon sampling and reciprocal sequence comparison[7] combined with Markov clustering[8] of complete proteomes to identify homology instead of one-way BLAST. We also define gene age based on occupancy in all taxa of a clade (see below) instead of using a single species as an anchor. While this is a robust methodology, it has some limitations seen in other BLAST-based analyses in that homology assignment might be affected by gene fusions, fissions, exon shuffling, lateral gene transfer, or presence of repetitive motifs[9]. We compared 62 genomes throughout Metazoa and their outgroups, including 44 genomes for 14 metazoan phyla (Fig. 1, Supplementary Fig. 2, Supplementary Data 1). These genomes were generated with different sequencing technologies, coverage, and gene prediction software. The completeness of these genomes was assessed using BUSCO[10] (Supplementary Data 1, Supplementary Fig. 3); 10 animal genomes showed percentages of missing genes above 15%, but five of them belong to animals that had suffered known genome reductions, while the others are within clades in which other genomes display very low values of missing genes in the BUSCO analyses. After performing all-vs-all BLAST, MCL

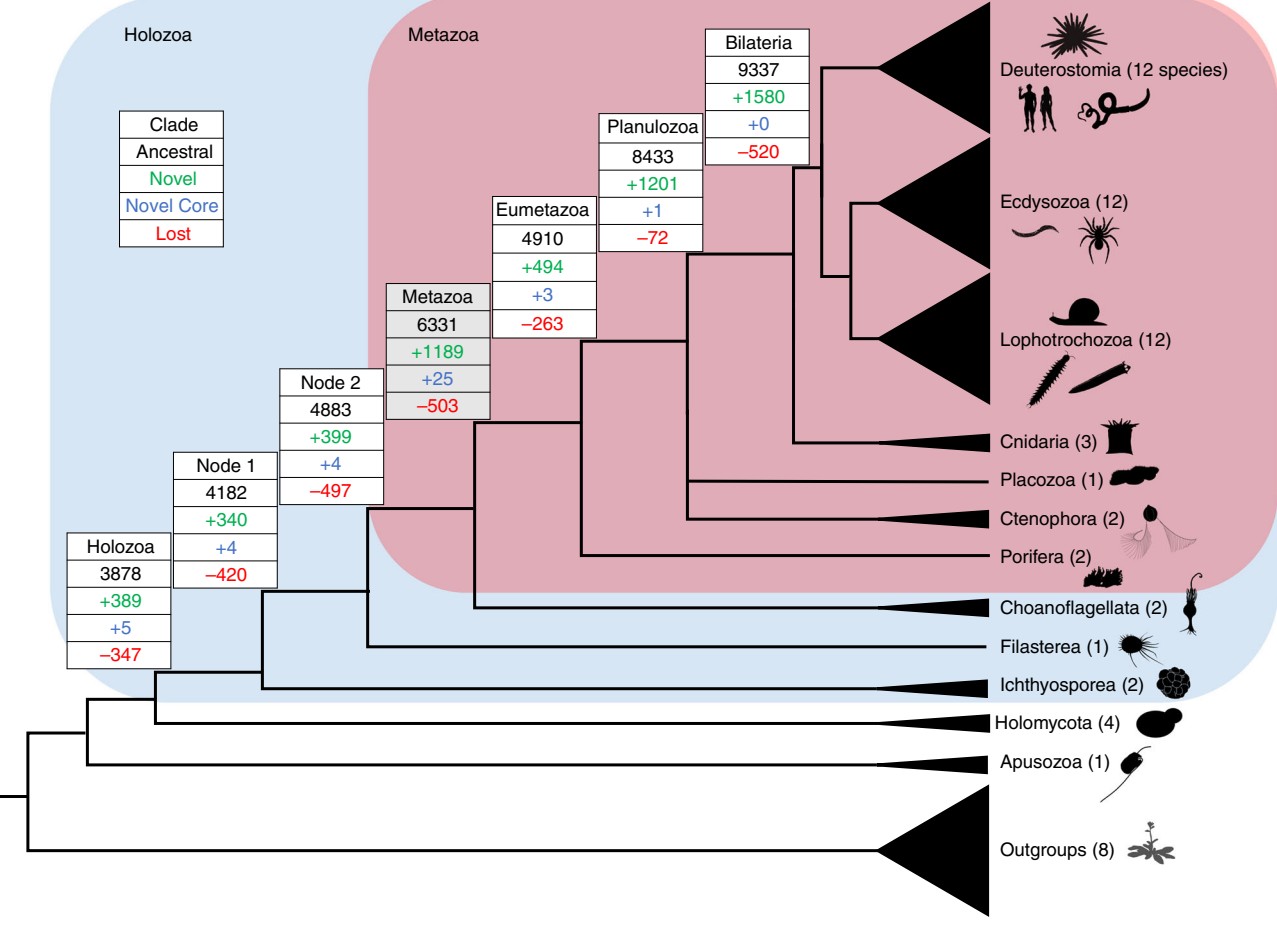

**Fig. 1** Reconstruction of ancestral genomes. Evolutionary relationships of the major groups included in his study[2]. Different categories of HG are indicated in each node, from top to bottom, Ancestral HG, Novel HG, Novel Core HG, and Lost HG. Values assume sponges as the sister group to other animals, and placozoans as sister group to Planulozoa (=Cnidaria + Bilateria); alternative phylogenetic hypotheses are explored in Supplementary Data 3-8. Organism outlines from phylopic.org and the authors

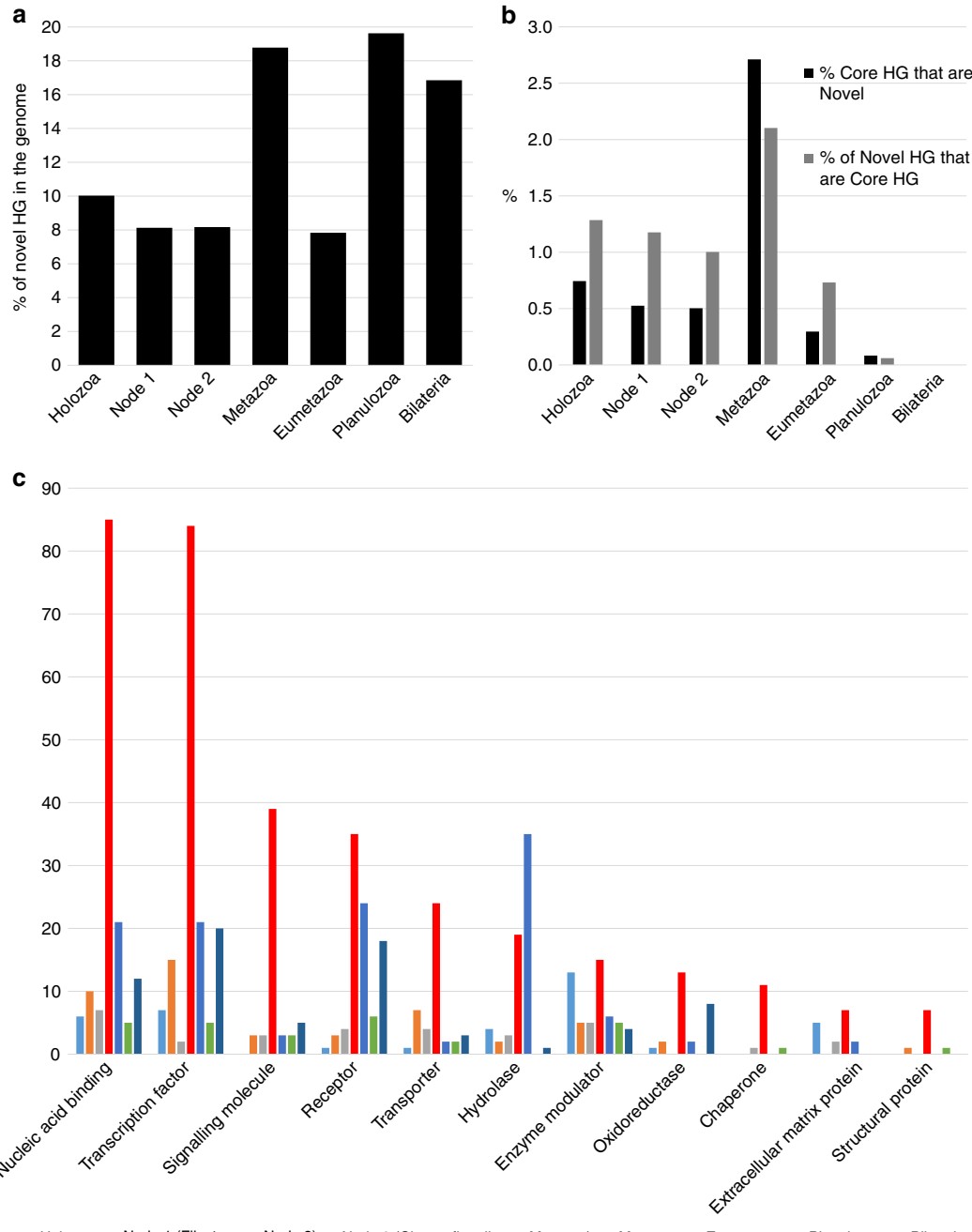

**Fig. 2** Novelty in ancestral genomes. **a** Proportion of Novel HG in the Ancestral HG for different holozoan ancestors. **b** Percentage of Core HG that are novel, and percentage of highly preserved genes among the Novel HG across different LCA. **c** Number of Protein Class GO hits for the fruit fly representatives of the Novel HG for the various phylogenetic nodes

displays similarities in two-dimensional space, within which the distance between proteins is proportional to the e-value of their BLAST comparison. MCL then applies graph theory and hidden Markov models to cluster sets of proteins based on their proximity in that space[8]. All 1,494,207 proteins were clustered in 268,440 homology groups (HG); all human genes are grouped within 9451 HG and fruit fly genes within 7618 HG (Supplementary Data [4]).

The inferred HG provide a contrast between the hierarchical nature of gene classifications used in the literature (e.g., gene families, classes, and superfamilies) and their evolutionary dynamics (clustering based on sequence divergence), drawing a

parallel with organisms' classical taxonomy and their phylogenetic relationships. A given HG can either include genes traditionally defined as gene families (e.g., *Iroquois* gene family, part of the TALE class, which is part of the homeobox genes superfamily[11]), as gene classes (e.g., POU class of the homeobox superfamily, containing multiple gene families), or as superfamilies (e.g., Wnt ligands, containing multiple gene classes that comprise diverse gene families). The recovery of traditional gene families/classes/superfamilies demonstrates the reliability of the clustering approach taken here. However, the evolutionary history of some genes is evident from their clustering; for example, even though the *Iroquois* gene family is classified within

**Table 1 Novel Core genes in the Animal Kingdom**

| Transcription factors | | Fruit fly genes examples | Human gene examples |
|---|---|---|---|
| Homeobox | NKL subclass, ANTP class | *tinman, distalless, ems* | *NKX2-5, DLX1, EMX1, VAX1, HLX, DBX1, BARX2* |
| | SIX class | *sine oculis, optix* | *SIX1, SIX6, ANHX* |
| | POU class | *pou proteins, nubbi, ventral veins lacking* | *POU1F1, POU2F1, POU6F1* |
| bHLH | hes/hairy | *hairy, hey, deadpan, clockwork orange* | *HES1, HES2, HEY1, HEYL* |
| | bHLH-PAS | *single minded, spineless, trachealess* | *EPAS1, HIF1A, SIM1, AHR, NPAS1* |
| | twist/hand | *twist, hand, target of Poxn, taxi, atonal* | *TWIST1, HAND1, SCX1, TCF15, PTF1A, NEUROD1, NEUROG1* |
| ETS | | *ets65A, anterior open, pointed* | *ETV4, ETS1, ELK1, ERG, FEV* |
| **Signalling pathways** | | | |
| Wnt | Wnt | *wingless* | *WNT1, WNT2, WNT3, WNT4, WNT10A* |
| | Frizzled | *frizzled, smoothened* | *SMO, FZD1, CORIN, SFRP2, FRZB* |
| | pangolin/TCF-LEF | *pangolin* | *TCF7, LEF1, TCF7L1* |
| | armadillo/beta-catenin | *armadillo* | *JUP, CTNNB1* |
| TGF-Beta | TGF-beta/BMP | *decapentaplegic, screw, activin beta* | *BMP2, BMP10, GDF1, INHBA, NODAL, TGFB1* |
| | SMAD | *mothers against decapentaplegic* | *SMAD1, SMAD2, SMAD3, SMAD9* |
| | TFG-beta receptor | *punt, saxophone, wishful thinking* | *TGFBR1, ACVR1B, AMHR2, BMPR1B* |
| | JNK pathway interaction | *sunday driver* | *MAPK8IP3, SPAG9* |
| **Transcripts polyadenilation** | | | |
| Cytoplasmic polyadenylation element binding protein (CPEB) | | *orb, orb2* | *CPEB1, CPEB* |
| **Cell adhesion** | | | |
| Fermitin | | *fermitin1 and 2* | *FERMT1, FERMT2* |
| Liprin | | *liprin alpha, beta, and gamma* | *PPFIA2, PPFIBP1* |
| Alpha-catenin | | *alpha-catenin* | *CTNNA1, CTNNA2* |
| **Cell cycle** | | | |
| RUN (after RaP2 interacting protein 8, UNC-14 and NESCA) | | *Dmel 0040348 CG3703, Dmel 0052461 CG32461* | *RUNDC1* |
| MAP kinase-activating death domain (MADD/GEF) | | *RAB3 GEF* | *MADD* |
| **Receptors** | | | |
| Nuclear hormone receptors | | *knirps, tailless, ecdysone receptor, seven up* | *RARB, RXRA, THRB, HNF4A, PPARG, NR4A1* |
| Neurotransmitter receptors | | *dopamine ecdysteroid receptor, serotonin receptor* | *HRH3, HTR2A, ABRA1D, DRD2, ADRA2B* |
| **Synaptic exocytosis** | | | |
| Calcium activated protein for secretion (CADPS) | | *calcium activated protein for secretion* | *CADPS, CADPS2* |
| Rab3-interacting molecules (RIM) | | *rim* | *RIMS1, RIMS2* |

List of the 25 novel HG that are highly retained in the genomes of the Animal Kingdom. Examples of modern gene families in bilaterians that belong to these HG are given for the fruit fly and human genomes; the specific named families were not necessarily present in the LCA of animals and may be product of later gene family expansions

the TALE class of the homeobox superfamily and emerged by duplication of a *TALE* gene, here it is clustered in a distinct HG separate from other *TALE* genes due to high sequence divergence. Hence, each HG defines a set of proteins that have distinctly diverged from others; it may contain one ortholog (gene family) or multiple paralogs (gene classes or superfamilies), regardless of their mechanism of origin and traditional classification. We believe this assignment is relevant from an evolutionary perspective as it highlights groups of proteins that are highly diverged from others. In contrast to approaches that atomize HG down to gene families based on further dissection of BLAST e-values (e.g., OrthoMCL[12]), these large clusters of orthologs and/ or paralogs are less prone to misassignments caused by

evolutionary patterns associated with gene family expansions, such as asymmetric evolution[13].

**Phylogenetic genomics**. A phylogenetically aware parsing script, which takes into consideration the evolutionary relationships between taxa together with the taxonomic occupancy of the HG, was written to reconstruct the HG present in each node. We assume that a given homology group can only emerge once, but may be lost multiple times. We extracted (a) 'Ancestral HG': all those present in the Last Common Ancestor (LCA) of a given clade, (b) 'Novel HG': those present in the LCA of a clade but not outgroups, (c) 'Novel Core HG': a subset of novel HG present in

every representative species within the clade, but permitting one absence, and (d) 'Lost HG': HG lost in the LCA of a clade (Fig. 1, Supplementary Data 2 to 8). The definition of 'Core' HG (Supplementary Note 1, Supplementary Data 2) allows for absence of the HG in one member of the ingroup to accommodate incompleteness of some genomes; ancestral lists of HG are therefore not affected by these genomes, as these HG are defined by their presence in at least two genomes belonging to the two main lineages within a clade. The Novel Core HGs were further validated by running BLASTP searches against the complete GenBank database (Methods section and Supplementary Data 9). We propose that the high level of retention of Novel Core genes is an indicator of functional importance. We placed poriferans as the sister group to all the other animal lineages, but alternative phylogenetic hypotheses[14,15] do not change the general patterns described below (Supplementary Data 3-8). The full list of gene IDs for each category of HG and different ancestors can be found in Supplementary Data 10.

**The genome of the Urmetazoon**. Our analyses infer the LCA of extant metazoans contained a total of 6331 Ancestral HG (4929 HG if ctenophores are sister group to metazoans, Supplementary Data 3 and 4). This will be below the total number of genes in the genome, since HG can contain multiple genes and some HG could have been lost from all surviving lineages. Tracing forward in time, human and fruit fly genomes have 13,034 and 8270 genes, respectively, belonging to these Ancestral HG; hence, 55–60% of human and fly protein-coding genes belong to HG present in the first animals. Interestingly, only 922 of these 6331 Ancestral HG are retained by 43 or 44 of the animal genomes, highlighting extensive gene loss or divergence across metazoan evolution. As the ancestral functions of these HG cannot be directly inspected, the functions of their descendants in modern genomes were used as a proxy. Gene ontology (GO) analyses[16] using fruit fly, which has extensive GO annotations, indicate that HG derived from the first animal genome were abundant in protein classes implicated in gene regulation (e.g., nucleic acid binding, transcription factors) and catalytic activities (e.g., hydrolases, transferases, and oxidoreductases; Supplementary Data 4). The most abundant GO pathways include many signalling pathways (e.g., Wnt, TGF-beta, cadherin signalling, integrin signalling; Supplementary Data 4). These findings are consistent with previous studies based on whole-genome comparisons[4,17,18]. At the level of both protein functional classes and pathways, there is considerable similarity between the ancestral animal genome and the extant genomes of fruit fly and human, and with those of successively earlier holozoan nodes (Supplementary Data 4).

**Genomic novelty in the origin of animals**. Concerning gene novelty, we infer the ancestral metazoan genome included a remarkable 1189 Novel HG; this number is similar to that inferred by previous study centred on the genome of a demosponge[19]. Our analyses indicate a threefold increase compared to novelties in the previous nodes (389, 340, and 399 novel HG in the older holozoan nodes; Fig. 1, Supplementary Data 3). The Novel HG comprise 19% of the total HG in the first metazoan, compared to only 8–10% in most older nodes examined; Planulozoa and Bilateria LCA nodes also have high proportions of Novel HG (Figs. 1, 2a, Supplementary Note 3, Supplementary Data 5). If ctenophores are considered sister group to the other animals, 782 Novel HG are recovered (Supplementary Data 3), a twofold novelty increase (16% of the HG in the genome, Supplementary Data 5). In addition, the metazoan LCA has a fivefold increase of Novel HG that are subsequently retained in all (or

almost all) descendent lineages (Fig. 2b, Supplementary Note 3, Supplementary Data 5), independently of the nature of the earliest animal lineage. The 1189 metazoan Novel HG contain a large number of regulatory functions compared to the metazoan Ancestral HG set (e.g., 23 vs 6% transcription factors, 11 vs 4% signalling), and are depauperate in enzymatic and metabolic functions (Supplementary Data 4 and 6). Comparing the Novel HG of each phylogenetic node, the number genes for several protein classes displays a peak in the animal ancestor (Fig. 2c); the novel functions which are more abundant in the LCA of animals (Supplementary Data 6) are nucleic acid binding (23%), transcription factors (23%), and signalling molecules (11%). Thus, the first animal genome was not only showing a higher proportion of Novel HG, but these also perform major multicellular functions in the modern fruit fly genome. The implication is that the transition was accompanied by an increase of genomic innovation, including many new, divergent, and subsequently ubiquitous genes encoding regulatory functions associated with animal multicellularity.

**Twenty five novel core groups of genes in animals**. We identified which novel gene functions were more retained through evolution of animals. We find a total of 25 Novel Core HG: protein groups emerging in the genome of the first animal and still present in at least 43 of the 44 metazoan genomes examined (Table 1); these are independent of alternative phylogenetic scenarios at the root of animals. For these 25 HG, we give examples of modern bilaterian gene families contained in these HG. Together they cover the spectrum of classical functions linked to animal multicellularity: gene regulation, signalling, cell adhesion and cell cycle. Earlier opisthokont LCA have much lower numbers of Novel Core HG (Fig. 1, Supplementary Data 3 and 7), supporting the importance of genetic innovation in the emergence of animals. Of the 25 HG, eight encode key components of two major signalling pathways: Wnt and TGF-Beta. Thus, for the Wnt pathway we find the ligand, a GPCR membrane receptor of the pathway (*frizzled*), a component of the system that translocates signal to the nucleus (*armadillo/beta-catenin*), and the downstream effector (*pangolin*). Both pathways have been hypothesized to be key to the transition to animal multicellularity, possibly through provision of an axial body patterning system[20]; their recovery confirms the validity of our approach to find biologically relevant genes. Seven of the Novel Core HG encode transcription factors (Table 1): NKL, SIX and POU homeodomain proteins (implicated in developmental decisions), Hes bHLH proteins (effectors of Notch signalling), bHLH-PAS proteins (responders to environmental and physiological signals), a bHLH group containing *twist*-related genes (cell lineage determination) and *Hand* (development of muscle and nervous systems), and a group of ETS genes (winged helix-turn-helix domain). Some, but not all, of these genes had previously been postulated as arising at the origin of Metazoa[21–30].

Several of the 25 Novel Core HG have not been previously implicated in the rise of metazoans; we suggest these overlooked proteins were equally important. They include CPEB proteins which bind the polyA tail of mRNA and activate or repress translation[31], adding a new layer to the regulation of gene expression in metazoans. Three other HG contain genes involved in cell adhesion, a key multicellular process; for example, fermitin and liprin are involved in activation of the integrin pathway and other cell-extracellular matrix interactions via focal adhesions[32]. The origin of these genes may have been pivotal to the origins of epithelia in poriferans and other Metazoa[33]. Two HG contain genes involved in cell cycle regulation, again critical in a multicellular organism: MADD and RUN. There is evidence that

MADD interacts with pathways linked to apoptosis[34], and the nematode orthologue is a guanine exchange factor (GEF)[35]. RUN is an effector of the Ras-like GTPase signalling pathway associated with cell proliferation[36]. Finally, there are four HG with functions related to the nervous system, striking considering our dataset includes animals with no nervous system (sponges and placozoans). These HG include neurotransmitter receptors, which had previously been associated with animal origins[18,37], but also CADPS and RIM proteins. CADPS and RIM regulate the fusion of vesicles with the cell membrane in neuroendocrine cells and in the presynaptic neurons, respectively[38]; both interact with soluble NSF-attachment protein receptors (SNARE), which are known for having undergone expansions in the origin of animals[3]. The study of those genes in early animal lineages could be important to understand the genesis of the nervous system.

**Gene losses**. Finally, 'Lost HG' (Fig. 1, Supplementary Data 2, 3, and 8) display a different temporal pattern to the ancestral and novel complements. The numbers of Lost HG are more consistent across nodes (347–520), with Metazoa and Bilateria showing the highest numbers (503 and 520, respectively). The exception is the Planulozoa LCA with 72 HG losses; however, inference of HG losses is sensitive to the identity of the immediate sister group of the clade of interest, and this is the area of the tree with highest uncertainty. The identification of the biological functions of 'Lost HG' (Supplementary Data 8) is hampered by their absence in model organisms, not only in the in-groups (e.g., humans, fruit fly, etc.) but also in key outgroups (e.g., *Arabidopsis thaliana*, *Saccharomyces cerevisiae*, etc.). The largest portion of annotated HG contains enzymes, including some kinases; developmentally relevant genes barely feature among the 'Lost HG', with few exceptions, such as Bromodomain containing transcription factors and WRKY genes in the LCA of Holozoa, or a Glutamate receptor in the origin of Bilateria (Supplementary Data 8). Two nodes with the highest values of both lost and novel HG are the Metazoa and Bilateria, which indicates that these transitions were associated with high turnover of genes and greater genomic plasticity.

## Discussion

The evolutionary genomics pipeline deployed here gives new insight into the genomes of our extinct ancestors. Our analyses show some striking similarities between the genomes of the first animal and other protist ancestors, copious in genes involved in gene regulation and metabolic activities. However, we also find a large number of gene novelties in the stem lineage of the Metazoa compared to other opisthokont ancestors, with the animal Novel Core HG having functions related to gene regulation, signalling, cell adhesion and cell cycle. Interestingly, the gross patterns of novelty do not change depending on the placement of sponges or comb jellies as sister group to all the other animals; absolute HG numbers are smaller in a 'ctenophores-first' scenario, which may be explained by the levels of gene loss in their genomes[39,40]. As in any comparative analysis the representation of taxa can have an impact, as observed by the discovery that some genes are older than previously thought when new holozoan genomes are sequenced[4]. In this study, although all genome data available were analysed, the inclusion of new genomes in the future may subtly change the reconstruction of HG present in different nodes. However, the independent validation of Novel Core HG using BLAST against GenBank gives confidence in the reliability of these placements. We stress the present study focusses on protein-coding genes, and it is possible that the evolution of non-coding genes, regulatory regions, and epigenetic mechanisms also played major roles in this transition[41].

There are two alternative scenarios that could explain these patterns depending on the length of the branch leading to the metazoan LCA. The first assumes that the birth rate of new genes was constant over time, thus the branch leading to the first metazoan was longer than other opisthokont internodes. This would imply an extended 'stew' in which the molecular components of animal biology were assembled. However, we note that molecular phylogenetic analyses do not generally show longer branches in the stem lineage of animals[2], contrary to this scenario. The second possibility involves many new genes emerging during a short 'popcorn' stage, caused either by a higher gene birth rate (perhaps produced by environmental factors elevating mutation rates, or due to whole-genome duplications), and/or a lower gene death rate (due to high integration of new genes into regulatory networks). In this scenario, the acquisition of multicellularity would quickly stabilise new molecular systems for cell adhesion, cell communication and the control of differential gene expression, as shown by the increase in proportion of Novel Core HG seen in the metazoan ancestor. These include genes previously hypothesized to be pivotal in the emergence of Metazoa, with additional genes singled out here for the first time as agents involved in the transition. This scenario is also consistent with enhanced rates of gene novelty in the ancestors of Planulozoa and Bilateria when embryonic patterning systems were being elaborated. Further data and analyses are needed to discriminate between the two scenarios.

While previous studies have shown gene gains at different nodes of the holozoan tree of life, here we reveal an increased number of novel HG along the branch leading to multicellular animals, highlighting the role of genome evolution in the origin of the Animal Kingdom.

## Methods

**Taxon sampling**. We designed a bioinformatics pipeline to identify homologous groups of proteins within and between genomes, and identify their phylogenetic origin (Supplementary Figure 1).

A broad wide genome sampling throughout Metazoa and their outgroups was compiled (Fig. 1, Supplementary Figure 2, Supplementary Data 1). We downloaded all predicted proteins (1,494,207) for the 62 eukaryotic genomes, including 44 genomes spanning 14 animal phyla, and 18 from 8 major eukaryotic outgroup lineages. Some of these genomes contain full gene annotations in their FASTA headers (indicated with an asterisk in Supplementary Figure 2). The completeness of all genomes was assessed using BUSCO[10] (Supplementary Data 1 and Supplementary Figure 3). Ten genomes are missing >15% of the genes in the BUSCO set, but these include high-quality genomes (indicated by their low percentage of fragmented genes) such as the two *Ciona* and the parasitic flatworm *Hymenolepis*, known to have suffered extensive gene losses[42–44]. Other genomes show higher levels of missing and fragmented genes (*Nematostella vectensis*, *Pinctada fucata*, *Mesobuthus martensii*, etc.), but they belong to clades in which other high-quality genomes are present.

**Homology assignment**. To infer the homology relationships between all genes, BLAST[7] searches were combined with a Markov clustering algorithm (MCL[8]). Best reciprocal BLAST searches compared the predicted proteins all versus all, resulting in a total 2,232,654,558,849 BLAST comparisons. Searches were performed with BLAST version 2.2.27+, using 10e−5 as the e-value threshold and tabular format as output. The BLAST output was then analysed with the version 11–294 of MCL, with the default granularity value ($I = 2.0$). MCL identified 268,440 probable gene homology groups (HG) across the dataset. Subdivision between orthologous and paralogous genes attempted in other methods (OrthoMCL) was not pursued, as the differing evolutionary dynamics of duplicates makes similarity-based approaches unsuitable for deducing gene duplications[45].

**Phylogenetically Aware Parsing Script**. A Phylogenetically-Aware Parsing Script (PAPS) was written in Perl to parse the output of MCL using a user-defined phylogenetic tree. The script includes the phylogenetic classification of the species of interest, which together with the taxonomic occupancy of the HG is used to reconstruct the presence of HG in different nodes of the evolutionary tree. The script has a user-friendly command-line interface which can be used to obtain lists of HG based on a set of criteria defined by the user, such as presence or absence of the HG in different clades at any phylogenetic level (from species to domain, Supplementary Fig. 2). This allows the user to tailor very flexible queries that are

processed by PAPS as required; for example, the program can produce lists of HG present in all Metazoa and absent in the rest of the taxa, or HG present in all Arthropoda minus one species, or HG present in Homo sapiens and at least 2 Hexapoda but absent in all Lophotrochozoa, etc. From these analyses, we extracted different sets of HG (Supplementary Note 1): 'Ancestral HG' (HG present in the Last Common Ancestor, or LCA, of a given clade), 'Novel HG' (HG present in the LCA of a clade but not any outgroups), the 'Core' subset for these two categories (subset of the HG present in every representative species within the clade, or absent only once), and 'Lost HG' (HG lost in the LCA of a clade). The different subsets of HG have different behaviours (Supplementary Note 1) and risks of false positives and negatives (Supplementary Note 2). Alternative phylogenetic hypotheses for the identity of the first metazoan and early nodes were explored (Supplementary Data 3 to 7).

The list of Novel Core HG for all the clades of interest (Fig. 1) was further tested, by performing BLASTP[7] searches against NCBI GenBank[46], which offers the broadest taxon sampling of molecular sequences available. For each of these HG, the fruit fly representatives were extracted and used as query in a local BLAST[47] search (default parameters). The search was executed against a local database containing all GenBank records (downloaded on 29 March 2017), excluding gene sequences from the in-groups with the option negative_gilist. The results can be found in Supplementary Data 9. Generally, the BLAST searches found very weak matches with poor e-values and/or low identity percentages (most under 30%), further validating the Novel Core HG obtained by combining BLAST, MCL, and our script. The full list of gene IDs for each category of HG and different ancestors can be found in Supplementary Data 10.

As discussed in further detail elsewhere (Supplementary Note 2), the likelihood of false positives and negatives is reduced by the fact that homology groups generally contain various gene families and multiple genes per genome (e.g., HG comprising Wnt genes ranges from three genes in each poriferan genome to 44 genes in chicken); therefore, all the representative genes of a given HG would need to be misassigned, missequenced, or misannotated in all the outgroup genomes to produce a false positive in lists of novelties, or in all the ingroup genomes to generate false negative, or in two or more genomes in lists of core genes (as absence in one genome is allowed in that case). As an example, urochordates and parasitic flatworms have lost up to 20–25 homeobox gene families[44,48], yet we still recover the HG to which those homeobox gene families belong because other members of these HG have not been lost in those genomes.

**Functional annotation of homology groups of genes**. To obtain a functional description of the lists of Ancestral HG and Novel HG for each opisthokontan node, their fruit fly genes were extracted and analysed in Panther GO11. The number of Gene Ontology (GO) hits were obtained for all the GO classification schemes: Protein Class, Molecular Function, Biological Process, Cellular Component, and Pathways. Graphics were produced in MS Excel, checking that the cell formatting was adequate to the values contained.

**Code availability**. The PAPS script is available in GitHub (https://github.com/PapsLab/Phylogenetic_Aware_Parsing_Script).

**Data availability**. The various sources for the genome data that support the findings of this study can be found in Supplementary Data 1.

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

## Acknowledgements

We thank Ignacio Maeso, Ferdinand Marlétaz, Sebastian M. Shimeld, Patrick Gemmell, and Thomas L. Dunwell for feedback during this project. Iñaki Ruiz-Trillo kindly provided genome data for different holozoan species. The authors are grateful to friends and colleagues who provided feedback on the manuscript. P.W.H.H. and J.P. acknowledge support from the European Research Council under the European Union's Seventh Framework Programme (FP7/2007-2013)/ERC grant [268513].

## Author contributions

J.P. and P.W.H.H. designed the study and analyses. J.P. performed the analyses. J.P. and P.W.H.H. wrote the manuscript.

## Additional information

**Competing interests:** The authors declare no competing interests.

