## [Peer Review File · Nature Communications]

Reviewers' comments:

Reviewer #1 (Remarks to the Author):

The manuscript by Paps and Holland examines a comprehensive set of animal genomes and representative outgroups to identify the homologous groups of genes whose appearance coincides with the origin of multicellular animals.

The manuscript is very clearly written and was very enjoyable to read. It is also thought-provoking and would be a very nice contribution to work in the field.

i have a few concerns / points, all of which would be straightforward to address:

1. in many places, the authors use terms like "first splitting" / "early splitting", etc. It would be better if they replaced those with scientifically more accurate terms like "sister group".

2. paragraph beginning on line 39: It would have been helpful if the authors here stated what model of HG gain / loss they used; i imagine they didn't allow for independent gain of HG and instead assumed that each HG was gained exactly once (or was ancestral). Some additional description would be helpful for understanding.

3. line 57: first animal genome - the "first" in this description is incorrect; i would favor replacing it with the scientifically valid and commonly used "Urmetazoan genome"...

4. lines 84-85: it took me a little bit to understand why the number of novel HG drops if ctenophores are assumed to be the sister group to the rest of the metazoans instead of sponges. May be worth adding a little bit more explanation so that the reader understands why that is the case? I imagine it has to do with the HG reconstruction along the phylogeny and the quite different HG content of sponge and ctenophore genomes...

5. My last point is perhaps the most important one and i believe it is essential that the authors address it. I did not see any discussion of caveats to the authors' (very nice) approach. A big caveat, in my mind, is that HG reconstruction depends quite heavily on the set of available genomes. Therefore, i think it would be wise for the authors to add a paragraph discussing this issue and how our ideas about genomic novelty might need (upward or downward) revision as more genomes from both metazoans as well as from non-metazoan close relatives are added. An example or two to illustrate the point would be useful as well. My own personal favorite has to do with collagen IV, which a recent study (<https://elifesciences.org/articles/24176>) showed was present only in animals and absent from the genomes of unicellular animal relatives. However, only a few months later another study (<https://elifesciences.org/articles/26036>) found collagen IV in Ministeria, suggesting that it likely predates animals. I don't think that's going to be the case for most HG the authors identify as novel, but it is certainly a possibility for some, so the authors should add some caution to the sampling-dependence of their results and inferences.

Reviewer #2 (Remarks to the Author):

This manuscript describes a new approach to date and classify gene families – homology groups – that is used to determine the evolutionary origin of metazoan genes and whether they are core to the metazoan genome (i.e. have been retained in over 95% of the genomes surveyed). This approach provides some new insights into the origin and evolution of metazoan gene families, substantiating a raft of earlier studies that either have focused on the evolution of a particular gene family, often transcription factors or signaling pathway members, or have used basal metazoan or non-metazoan holozoan genomes to infer ancestral conditions. Indeed, the results present here match very closely those presented in Srivastava et al. [Nature (2010) 466:720].

The approach presented in this manuscript appears to have potential to overcome some the deficiencies present in comparable methods, namely Ortho MCL and phylostratigraphy. The use of an 'all against all' reciprocal sequence comparison combined with Markov clustering of complete proteomes from a large number of taxa is simple and appears to have widespread utility.

In addition, the classification of homology groups into ancestral vs novel, and core vs non-core is useful in determining if gene families evolved in concert with a particular evolutionary event or were co-opted from a more ancient role. In this paper, these events are related to the evolution of metazoans. This seems like useful way to describe genes.

However, the results presented in this manuscript do not appear to add markedly to existing knowledge about the evolution of particular gene families; that said, there are some new and important additions. Importantly, this homology-based approach does not seem to increase the quality of gene family assignment when compared to Ortho MCL and phylostratigraphy. Indeed, some of limitations of these two former methods in resolving gene families and classes to a level that is relevant to the origin and evolution of animals appears to also exist in this new method. For instance, amongst the 25 novel core HGs inferred to be present in the metazoan LCA, there appear to be a number of incorrect assignments. For instance, nuclear receptor Group 4 and the bHLH Twist are both assigned the metazoan LCA. Numerous publications from sponges, ctenophores and non-metazoan holozoans provide no support for these assertions; NR2 appears to be the only group present in sponges and ctenophores, and there are bHLH genes loosely related to Twist in the basal metazoan survey in this study but no Twist (the calcisponge *Sycon*, which was not included in this study, does not have Twist but has more closely related bHLH members). This homology based method also appears to struggle with identifying/assigning gene families comprised of repeating and/or complex domain architectures (e.g. those typifying many putative innate immunity gene families; see Table S8).

In summary, it appears this high-throughput approach and the associated gene classification system has potential to contribute to our understanding of genome evolution.

However, it appears that further optimisation and 'ground-truthing' is necessary before publication. Minimally, this approach should support previous studies, many which have not been cited in this manuscript, that have a detailed focus on the evolution of particular gene families. In addition to this general concern, some of the supplementary tables (e.g. S6 – S8) are difficult to interpret. Finally, it is important to include a full list of novel and ancestral HGs at each of the nodes.

Reviewer #3 (Remarks to the Author):

What are the major claims of the paper?

The manuscript infers the number of protein clusters (termed homology groups) in the extinct ancestors of opisthokonts, metazoa, eumetazoa, bilateria etc. and calculates how many of these protein groups are either new or not, or have been lost compared to the preceding ancestor. The main claim is that the ancestral metazoan genome has undergone a multifold increase in the number of novel homology groups compared to the other hypothetical ancestors. Of the ~1,200 novel groups, ~25 have been retained in nearly all contemporary metazoans. Five of these metazoan-wide retained, new homology groups had not been associated before with the emergence of animals. The potential biological role of these groups is discussed.

Are they novel and will they be of interest to others in the community and the wider field?

An original aspect of the work is its focus on NOVEL metazoan gene groups. Previous studies have focused more on metazoan genes that pre-dated multicellularity. The approach developed for comparing ancestral gene complements highlights the challenges of this task. Interesting for the wider scientific community is the topic --emergence of animals--, to which most of us can easily relate.

Is the work convincing, and if not, what further evidence would be required to strengthen the conclusions?

A number of ambiguities, shortcomings and open questions reduce the persuasiveness of the manuscript. First, the meaning of 'genomic novelty' is vaguely defined. After studying carefully the Supp. Mat. and reading cited literature, it appears that it is essentially defined as the occurrence of new groups of proteins that happen to cluster together via the MCL algorithm. This kind of association can mean very different things biologically: proteins already present in predecessors, but whose sequence diverged to a degree that they are not retained in the original cluster at the given parameter setting; gene duplication followed by accelerated sequence evolution; horizontal gene transfer.

With the above in mind, it is confusing to read the claim that the PAPS script would infer the 'evolutionary origin' of homology groups (see e.g. manuscript L. 24, 40; Suppl Mat. L. 42-

43).

In addition, one of the central conclusions appears to be incorrect, i.e. the claimed 'increase of genomic novelty [specifically] in the dawn of the Metazoa' (Abstract). The 3-fold increase (Lines 78-79) is probably due to a biased comparison, because the nodes predating metazoa contain only <10% non-metazoa. Had more non-metazoan genomes been used in (or were available for) the analysis, then several the homologous groups considered novel in metazoa would have most likely shown up as already present in non-metazoan holozoa or opisthokonts, thus reducing the number of novel metazoan HGs. A statistical consideration and evaluation would be needed here.

Further, a doubt emerges whether the taken approach is fully valid. One issue is the selection of the earliest-splitting branch in a given clade (together with one of the later offshoots) for inferring the set of ancestral homology groups. It would be more meaningful in a phylogenetic sense to choose among the basally-diverging taxa one that is least divergent, i.e. has the shortest branch. This would likely broaden the predicted ancestral homology-group set, and thus reduce the number of novel groups inferred. Similarly, predicted novel metazoan-specific groups may not have been thoroughly validated (the exact procedure is not specified; it should be done by TBLASTN against NCBI nr), which would result again in an overestimated number of novel metazoan homology groups. In addition, the validation of novelty uses the Drosophila instance of a protein group for Blast queries against all other organisms (Suppl. Mat. L61 etc). Has it been tested whether the Drosophila sequence would pick up all homologs that would be found when using the other members of the group as a query? Again, missed homologs in non-metazoa would lower the number of novel groups.

Confusing is also that the numbers of ancestral, novel, and lost groups appear arithmetically not consistent when comparing neighbor nodes. How is it possible that the nr. of ancestral groups in Metazoa (6,331) is larger than the total nr. of groups in the preceding Node 2 (4,883+399) ?

Finally, the validity of the in silico-inferred homology groups should be shown at least for one case, by expert validation of the predicted members in a given group.

Will the paper influence thinking in the field?

Yes, because there is no consensus about how to do (efficiently) across- and along-tree genome comparisons.

Further concerns and suggestions

In sum, Nature Communications with its narrow page limit, seem not to be the optimal choice for such a relatively complex topic. The central concepts are too shortly treated in the article to be fully comprehensible. In fact, the methodology, which is a central part of the work, should be presented and justified in the article itself, and not be predominantly put into the Suppl. Material as 'just a detail'.

Also, it would be interesting to think about hypotheses what gains and losses to expect

during the transition to multicellularity. This could help to better structure the finding (L.70, 89 to end).

Minor points

Article manuscript

- The characterization of Metazoa as the 'largest' multicellular eukaryotic kingdom (Line 10) is ambiguous. First, what exactly is a 'kingdom', i.e. which groups are being compared? Second, what means 'largest'? Is it the nr. of described species or predicted species (e.g. based on metagenomics) or else, especially as the species concept is not coherent across eukaryotes.
 - L. 35, what means 'high quality'?
 - The pipeline relies on protein sets predicted by others and different methodologies. The potential consequences of this inconsistency should be mentioned somewhere.
- Supp. Mat. (as mentioned above, much of the information in Supp. Mat. would be much better suited for the article part):
- There are a number of typos in the text and figures (e.g. Capsaspora owc...)
 - It would be informative for the reader if the main difference between PAPS and OrthoMCL were detailed.
 - A short description of the MCL algorithm would help to appreciate the appropriateness of the chosen approach. For example, does MCL consider the length of identity?
 - Fig. S2, "Chromalveolata" is an obsolete name (it implies that the common ancestor once had plastids, which is highly contentious).

Reviewer #1 (Remarks to the Author):

The manuscript by Paps and Holland examines a comprehensive set of animal genomes and representative outgroups to identify the homologous groups of genes whose appearance coincides with the origin of multicellular animals. The manuscript is very clearly written and was very enjoyable to read. It is also thought-provoking and would be a very nice contribution to work in the field.

We are pleased to read that Reviewer 1 finds the paper clear and interesting.

I have a few concerns / points, all of which would be straightforward to address:

1. in many places, the authors use terms like "first splitting" / "early splitting", etc. It would be better if they replaced those with scientifically more accurate terms like "sister group".

These terms have now been replaced following the referee's suggestion.

2. paragraph beginning on line 39: It would have been helpful if the authors here stated what model of HG gain / loss they used; i imagine they didn't allow for independent gain of HG and instead assumed that each HG was gained exactly once (or was ancestral). Some additional description would be helpful for understanding.

We agree with the referee, we now explain the assumptions used in that section (line 80).

3. line 57: first animal genome - the "first" in this description is incorrect; i would favor replacing it with the scientifically valid and commonly used "Urmetazoan genome"...

This term has now been replaced, Urmetazoon is now present in the section header (line 100).

4. lines 84-85: it took me a little bit to understand why the number of novel HG drops if ctenophores are assumed to be the sister group to the rest of the metazoans instead of sponges. May be worth adding a little bit more explanation so that the reader understands why that is the case? I imagine it has to do with the HG reconstruction along the phylogeny and the quite different HG content of sponge and ctenophore genomes...

This is a very interesting point. While the overall patterns do not change when ctenophores are placed as sister to the rest of animals, the novel HG numbers are smaller (except the subset of "core" numbers, which are not affected by the phylogeny of the ingroup). We believe this is due to

the gene loss reported in comb jellies, which has no effect under a 'sponges-first' scenario. We have now added a comment in the Discussion section (line 222).

5. My last point is perhaps the most important one and i believe it is essential that the authors address it. I did not see any discussion of caveats to the authors' (very nice) approach. A big caveat, in my mind, is that HG reconstruction depends quite heavily on the set of available genomes. Therefore, i think it would be wise for the authors to add a paragraph discussing this issue and how our ideas about genomic novelty might need (upward or downward) revision as more genomes from both metazoans as well as from non-metazoan close relatives are added. An example or two to illustrate the point would be useful as well. My own personal favorite has to do with collagen IV, which a recent study (<https://elifesciences.org/articles/24176>) showed was present only in animals and absent from the genomes of unicellular animal relatives. However, only a few months later another study (<https://elifesciences.org/articles/26036>) found collagen IV in Ministeria, suggesting that it likely predates animals. I don't think that's going to be the case for most HG the authors identify as novel, but it is certainly a possibility for some, so the authors should add some caution to the sampling-dependence of their results and inferences.

This is an important remark, we have added a comment in the Discussions section together with other limitations of our approach (line 226).

Reviewer #2 (Remarks to the Author):

This manuscript describes a new approach to date and classify gene families – homology groups – that is used to determine the evolutionary origin of metazoan genes and whether they are core to the metazoan genome (i.e. have been retained in over 95% of the genomes surveyed). This approach provides some new insights into the origin and evolution of metazoan gene families, substantiating a raft of earlier studies that either have focused on the evolution of a particular gene family, often transcription factors or signaling pathway members, or have used basal metazoan or non-metazoan holozoan genomes to infer ancestral conditions. Indeed, the results present here match very closely those presented in Srivastava et al. [Nature (2010) 466:720].

The approach presented in this manuscript appears to have potential to overcome some the deficiencies present in comparable methods, namely Ortho MCL and phylostratigraphy. The use of an ‘all against all’ reciprocal sequence comparison combined with Markov clustering of complete proteomes from a large number of taxa is simple and appears to have widespread utility.

In addition, the classification of homology groups into ancestral vs novel, and core vs non-core is useful in determining if gene families evolved in concert with a particular evolutionary event or were co-opted from a more ancient role. In this paper, these events are related to the evolution of metazoans. This seems like useful way to describe genes.

We appreciate the positive comments of Reviewer 2.

However, the results presented in this manuscript do not appear to add markedly to existing knowledge about the evolution of particular gene families; that said, there are some new and important additions. Importantly, this homology-based approach does not seem to increase the quality of gene family assignment when compared to Ortho MCL and phylostratigraphy. Indeed, some of limitations of these two former methods in resolving gene families and classes to a level that is relevant to the origin and evolution of animals appears to also exist in this new method. For instance, amongst the 25 novel core HGs inferred to be present in the metazoan LCA, there appear to be a number of incorrect assignments. For instance, nuclear receptor Group 4 and the bHLH Twist are both assigned the metazoan LCA. Numerous publications from sponges, ctenophores and non-metazoan holozoans provide no support for these assertions; NR2 appears to be the only group present in sponges and ctenophores, and there are bHLH genes loosely related to Twist in

the basal metazoan survey in this study but no Twist (the calcisponge Sycon, which was not included in this study, does not have Twist but has more closely related bHLH members).

We agree that the study makes ‘some new and important additions’, but disagree with the suggestion that our approach has comparable accuracy to Ortho MCL and phylostratigraphy. The examples given by the reviewer, concerning specific NHR and bHLH genes are interesting. We understand the concern, but we believe it was caused by some imprecise wording in our original manuscript. Homology groups are hierarchical and most contain multiple gene families, but this doesn’t mean that all the extant families contained within a homology group were present in the Urmetazoan genome. In the Nuclear Hormone Receptor (NHR) example, the “founder” gene of the NHR homology group was present in the LCA of all extant animals, but some NHR families contained in that HG are specific to vertebrates or mammals (or other inclusive clades) and thus absent in the animal LCA. We hope we have now explained this better in the text (line 156) and Table 1 legend, and thank the reviewer for drawing our attention to the confusion.

This homology based method also appears to struggle with identifying/assigning gene families comprised of repeating and/or complex domain architectures (e.g. those typifying many putative innate immunity gene families; see Table S8).

The reviewer highlights an important limitation of all BLAST-based approaches. These and other problems with BLAST have been added to the text (line 32). We also highlighted other general limitations (focus on protein-coding genes) of our approach in the Discussion section (lines 227-235).

In summary, it appears this high-throughput approach and the associated gene classification system has potential to contribute to our understanding of genome evolution. However, it appears that further optimisation and ‘ground-truthing’ is necessary before publication. Minimally, this approach should support previous studies, many which have not been cited in this manuscript, that have a detailed focus on the evolution of particular gene families.

We agree that some references have been neglected, due to the previous shorter format of the manuscript. This oversight has now been corrected and we took advantage of the advice of the editor of referees to expand the text, and the bibliography has been expanded from 22 to 40 references.

In addition to this general concern, some of the supplementary tables (e.g. S6 –S8) are difficult to interpret. Finally, it is important to include a full list of novel and ancestral HGs at each of the nodes.

We appreciate the referee's feedback, the Table descriptions have been expanded (Supp Info lines 299-311); we hope this makes easier to follow them. We have provided now the full list of all the genes included in all HG in the new Supplementary File S2.

Reviewer #3 (Remarks to the Author):

What are the major claims of the paper?

The manuscript infers the number of protein clusters (termed homology groups) in the extinct ancestors of opisthokonts, metazoa, eumetazoa, bilateria etc. and calculates how many of these protein groups are either new or not, or have been lost compared to the preceding ancestor. The main claim is that the ancestral metazoan genome has undergone a multifold increase in the number of novel homology groups compared to the other hypothetical ancestors. Of the ~1,200 novel groups, ~25 have been retained in nearly all contemporary metazoans. Five of these metazoan-wide retained, new homology groups had not been associated before with the emergence of animals. The potential biological role of these groups is discussed.

Are they novel and will they be of interest to others in the community and the wider field?

An original aspect of the work is its focus on NOVEL metazoan gene groups. Previous studies have focused more on metazoan genes that pre-dated multicellularity. The approach developed for comparing ancestral gene complements highlights the challenges of this task. Interesting for the wider scientific community is the topic --emergence of animals--, to which most of us can easily relate.

We are glad to hear that the referee finds the manuscript interesting; the discovery that there were so many novel metazoan gene groups is also the new finding that excites us.

Is the work convincing, and if not, what further evidence would be required to strengthen the conclusions?

A number of ambiguities, shortcomings and open questions reduce the persuasiveness of the manuscript. First, the meaning of 'genomic novelty' is vaguely defined. After studying carefully the Supp. Mat. and reading cited literature, it appears that it is essentially defined as the occurrence of new groups of proteins that happen to cluster together via the MCL algorithm. This kind of association can mean very different things biologically: proteins already present in predecessors, but whose sequence diverged to a degree that they are not retained in the original cluster at the given parameter setting; gene duplication followed by accelerated sequence evolution; horizontal gene transfer. With the above in mind, it is confusing to read the claim that the PAPS script would infer the 'evolutionary origin' of homology groups (see e.g. manuscript L. 24, 40; Suppl Mat. L. 42-43).

We thought we had been clear with our definition, since we think that ultimately novelty/divergence of amino acid sequence is more important to biological function than is precise mode of origin. To hopefully assuage the reviewer's concern, we have expanded the definition of HG in the manuscript using addition of new text and using parts of the Supplementary Material, describing the processes detailed by the referee (lines 52-74). We have also reworded the explanation of the PAPS script (main text, lines 77-20; Supp Info, lines 40-44).

In addition, one of the central conclusions appears to be incorrect, i.e. the claimed 'increase of genomic novelty [specifically] in the dawn of the Metazoa' (Abstract). The 3-fold increase (Lines 78-79) is probably due to a biased comparison, because the nodes predating metazoa contain only <10% non-metazoa. Had more non-metazoan genomes been used in (or were available for) the analysis, then several the homologous groups considered novel in metazoa would have most likely shown up as already present in non-metazoan holozoa or opisthokonts, thus reducing the number of novel metazoan HGs. A statistical consideration and evaluation would be needed here.

We used all the genomic data available. As discussed with Reviewer 1, taxon sampling is always relevant. The addition of new genomes could conceivably change some very specific results, but (like Reviewer 1) we believe it will not change the overall patterns observed here. Since this is an important point, we have now indicated this in the text (line 226).

To address the specific point, we believe there are no biased comparisons in our analyses. First, all the nodes in our phylogeny (Figure 1), have immediate sister groups composed of a low number of representatives compared to the ingroup, including the ones after the origins of animals that contain few but well-sequenced genomes (e.g. 42 Eumetazoa vs 2 Porifera). If sampling inequalities elevate the measured number of novelties, then we should expect to see such bias in all nodes, yet only the Metazoa shows an increase of novelty.

Second, the inference of novelty is not done only against the immediate outgroup (e.g. 44 Metazoa genomes vs 2 choanoflagellates), but against all the outgroups (e.g. 44 Metazoa vs 18 outgroups). The only novel genes that can be affected by the immediate outgroup are genes that emerged in the LCA of the ingroup and the immediate outgroup but were lost in all the members of the outgroups (e.g. emerged in the LCA of choanoflagellates and metazoans, but lost in choanoflagellates); this possibility is discussed in the Supplementary Material.

Finally, and very importantly, each inference of core novelty from the genomic data is then tested by broader BLAST searches against all GenBank (see below). Specifically, we took all Novel

Core HG in all nodes of our phylogeny, and searched these against all GenBank; not once did this recover a false positive, thus given us confidence in the values observed. Given that some of the Novel Core genes comprise HG with multiple families from well-studied developmental genes, with related genes validated by the BLAST search, it would be surprising that they had been never been captured in any study in the literature (including genomes, transcriptomes, and PCR-based analyses). We believe opting to use HG as defined by MCL instead of trying to define gene families/orthologs (OrthoMCL, InParanoid, OMA) reduces false positives and gives results with more evolutionary meaning.

Further, a doubt emerges whether the taken approach is fully valid. One issue is the selection of the earliest-splitting branch in a given clade (together with one of the later offshoots) for inferring the set of ancestral homology groups. It would be more meaningful in a phylogenetic sense to choose among the basally-diverging taxa one that is least divergent, i.e. has the shortest branch. This would likely broaden the predicted ancestral homology-group set, and thus reduce the number of novel groups inferred.

We used all the genomes for all the species available when we started this project, thus we couldn't select any shortest representatives. The key organisms in this study are not traditionally considered long-branched taxa (except ctenophores, where transcriptome-based data show the whole group is long-branched).

Similarly, predicted novel metazoan-specific groups may not have been thoroughly validated (the exact procedure is not specified; it should be done by TBLASTN against NCBI nr), which would result again in an overestimated number of novel metazoan homology groups.

We agree with the reviewer that a protein-based BLAST constitutes a very strong validation test, and the two protein-based options are TBLASTN and BLASTP. Usually TBLASN is used when there is the suspicion that a particular gene has not been annotated in a single target genome, we have used profusely in the past to annotate homeobox genes in new genomes. But in the current scenario, our queries comprise multiple genes and gene families (e.g. 23 *Drosophila* sequences for the NK Class homeobox genes, 21 for Nuclear Hormone Receptors, etc). And the target database is not a single genome, but the whole GenBank database with its vast genetic and taxonomic wealth (coming from genomes, transcriptomes, and PCR-based studies) and a large number of outgroup species for each of our nodes. We think BLASTp is the right approach in this case, as it is very unlikely that for a given Novel Core HG (that contains multiple genes and/or gene families) all its

representatives have been missed in all the gene annotations of all the genomes (and PCR-based studies) for the large number of outgroups available in NCBI.

We would like to stress that this test using BLASTP to validate the Novel Core HG is highly original and also time consuming, as compiling the negative-GI list to exclude the ingroups from the BLAST search is not straightforward. This is a validation test not seen in most comparative genomics papers, and we hope reassures the reviewer and editor of our commitment to deliver reliable results.

In addition, the validation of novelty uses the *Drosophila* instance of a protein group for Blast queries against all other organisms (Suppl. Mat. L61 etc). Has it been tested whether the *Drosophila* sequence would pick up all homologs that would be found when using the other members of the group as a query? Again, missed homologs in non-metazoa would lower the number of novel groups.

Regarding the use of *Drosophila*, we checked first if we were achieving the same results using human sequences for a random selection of HG. We did not detect any difference, and considering these BLAST searches are not computationally trivial, we opted to use *Drosophila* for the rest of the BLAST searches. As remarked above, most HG have multiple paralogs in multiple organisms (ingroups and outgroups), so for most HG we are using multiple genes as query, and expecting to find multiple hits in multiple outgroup genomes. In none of the 42 accumulated Novel Core HG genes have we found a hit in outgroups.

Some of our novel HG are well-known gene families related to development that previous studies already proved to be animal-specific, some performed by the authors of this study (e.g. subgroups of homeobox genes). The fact that the Novel Core HG are supported by independent studies performed by other researchers is an independent validation of these HG.

Confusing is also that the numbers of ancestral, novel, and lost groups appear arithmetically not consistent when comparing neighbor nodes. How is it possible that the nr. of ancestral groups in Metazoa (6,331) is larger than the total nr. of groups in the preceding Node 2 (4,883+399) ?

The numbers in consecutive nodes are not additive, mostly due to independent gene losses in different lineages. For example, the 4,883 Ancestral HG in Node 2 (Metazoa + Choanoflagellata) are HG present in at least one of the choanoflagellates and at least one metazoan; and example might be genes present in the choanoflagellates and eumetazoans, but absent in sponges. The Metazoan

Ancestral HG (6331 HG) have to be present in at least one sponge and at least one eumetazoan, therefore the genes from the previous example wouldn't be present in metazoan ancestral complement because they have been lost in sponges.

Finally, the validity of the in silico-inferred homology groups should be shown at least for one case, by expert validation of the predicted members in a given group.

We agree with the referee, and so we have added a section describing an example from homeobox genes (line 54). Similarly, many of the Novel Core HG perfectly match classical gene families/classes/superfamilies supported by the literature, validating the results of the MCL clustering.

Will the paper influence thinking in the field? Yes, because there is no consensus about how to do (efficiently) across- and along-tree genome comparisons.

Further concerns and suggestions

In sum, Nature Communications with its narrow page limit, seem not to be the optimal choice for such a relatively complex topic. The central concepts are too shortly treated in the article to be fully comprehensible. In fact, the methodology, which is a central part of the work, should be presented and justified in the article itself, and not be predominantly put into the Suppl. Material as 'just a detail'. Also, it would be interesting to think about hypotheses what gains and losses to expect during the transition to multicellularity. This could help to better structure the finding (L.70, 89 to end).

We appreciate the referee feedback, and we have now expanded the length of the manuscript to better describe concepts that were not clear before. Following the advice, we also reworded the Introduction and the Discussion to introduce hypotheses about which biological functions are expected to be seen in such transition.

Minor points

Article manuscript

- The characterization of Metazoa as the 'largest' multicellular eukaryotic kingdom (Line 10) is ambiguous. First, what exactly is a 'kingdom', i.e. which groups are being compared? Second,

what means 'largest'? Is it the nr. of described species or predicted species (e.g. based on metagenomics) or else, especially as the species concept is not coherent across eukaryotes.

This sentence has been reworded (line 11).

- L. 35, what means 'high quality'?

This sentence has been reworded (line 44).

- The pipeline relies on protein sets predicted by others and different methodologies. The potential consequences of this inconsistency should mentioned somewhere.

This is highlighted in the text now (line 38).

Supp. Mat. (as mentioned above, much of the information in Supp. Mat. would be much better suited for the article part):

We agree and appreciate this comment, and we have followed the advice.

- There are a number of typos in the text and figures (e.g. Capsaspora owc...)

We apologize, but couldn't find a typo in the name of *Capsaspora owczarzaki*, but we are happy to correct them if others are discovered.

- It would be informative for the reader if the main difference between PAPS and OrthoMCL were detailed.

We appreciate the comment of the referee, this has been added to the manuscript now (line 69).

- A short description of the MCL algorithm would help to appreciate the appropriateness of the chosen approach. For example, does MCL consider the length of identity?

We agree, this has been added to the manuscript now (line 45).

- Fig. S2, "Chromalveolata" is an obsolete name (it implies that the common ancestor once had plastids, which is highly contentious).

We removed the Chromoalveolata designation.

REVIEWERS' COMMENTS:

Reviewer #1 (Remarks to the Author):

The authors have satisfactorily addressed my comments and concerns.

Reviewer #2 (Remarks to the Author):

The revised manuscript is improved and has addressed most concerns. The only point that should be addressed is the lack of recognition that the "remarkable 1189 Novel HG" (line 127) they detected in the ancestral metazoan genome is not new. Srivastava et al. (2010. Nature 466:720) noted "Out of 4,670 pan-metazoan gene families defined by clustering sponge and eumetazoan peptides, 1,286 (27%) seem to be metazoan-specific". It is worth noting the consistency in the number of metazoan novel HGs at around 1200 between this and the earlier study.

Minor points

The use of "Planulozoa" and "Eumetazoa" – the former needs to be defined and the latter only applies if sponges, and not ctenophores, are sister to all other animals. These probably need to be explained somewhere.

Line 41 Supp – what is meant by "applying evolutionary thinking"?

Line 83 Supp – "As an example, urochordates and parasitic flatworms have lost up to 20- 25 homeobox gene families, yet we still recover the HG to which those homeobox gene families belong because other members of these families have not been lost in those genomes." Probably needs to be reworded. How can these taxa lost these homeobox families yet have other members of the families?

Make sure terminology used in the Supp is consistent with the main text (e.g. "first-splitting metazoans").

Reviewer #3 (Remarks to the Author):

I recommend publication of the revised manuscript version

Reviewer #1 (Remarks to the Author):

The authors have satisfactorily addressed my comments and concerns.

We are glad to have suitably addressed the suggestions of this referee.

Reviewer #2 (Remarks to the Author):

The revised manuscript is improved and has addressed most concerns. The only point that should be addressed is the lack of recognition that the “remarkable 1189 Novel HG” (line 127) they detected in the ancestral metazoan genome is not new. Srivastava et al. (2010. Nature 466:720) noted “Out of 4,670 pan-metazoan gene families defined by clustering sponge and eumetazoan peptides, 1,286 (27%) seem to be metazoan-specific”. It is worth noting the consistency in the number of metazoan novel HGs at around 1200 between this and the earlier study.

The consistency with the Srivastava analysis (which used fewer genomes) is remarkable. We now cite that important genome paper at the appropriate place.

Minor points

The use of “Planulozoa” and “Eumetazoa” – the former needs to be defined and the latter only applies if sponges, and not ctenophores, are sister to all other animals. These probably need to be explained somewhere.

Thank you. We now define Planulozoa in the figure legend. We do not use the term Eumetazoa in the main text, and where we used it in the Supplementary we have now defined or rephrased

Line 41 Supp – what is meant by “applying evolutionary thinking”?

We agree this wording was loose, and we have rephrased to “using a user-defined phylogenetic tree”

Line 83 Supp – “As an example, urochordates and parasitic flatworms have lost up to 20- 25 homeobox gene families, yet we still recover the HG to which those homeobox gene families belong because other members of these families have not been lost in those genomes.” Probably needs to be reworded. How can these taxa lost these homeobox families yet have other members of the families?

Thank you spotting this typo. We have corrected to “other members of these HG have not been lost”

Make sure terminology used in the Supp is consistent with the main text (e.g. “first-splitting metazoans”).

Thank you for noting the inconsistency. We have now rephrased all such instances in the Supplementary to ‘sister group’ etc.

Reviewer #3 (Remarks to the Author):

I recommend publication of the revised manuscript version

We agree with this referee’s suggestion.